# Uncovering shared and tissue-specific molecular adaptations to intermittent fasting in liver, brain, and muscle

Yibo Fan[1,2†], Senuri De Silva[3,4†], Nishat I Tabassum[1,2], Xiangyuan Peng[1,2], Vernise JT Lim[1,2], Xiangru Cheng[1,2], Keshava K Datta[5], Rohan Lowe[5], Terrance G Johns[6], Mark P Mattson[7], Suresh Mathivanan[2], Christopher G Sobey[1,8,9], Eitan Okun[10], Yong U Liu[11], Guobing Chen[12], Mitchell Kim Peng Lai[13], Dong-Gyu Jo[14], Jayantha Gunaratne[3,4*], Thiruma V Arumugam[1,2,14*]

[1]Department of Microbiology, Anatomy, Physiology and Pharmacology, School of Agriculture, Biomedicine and Environment, La Trobe University, Melbourne, Australia; [2]La Trobe Institute for Molecular Science, La Trobe University, Melbourne, Australia; [3]Translational Biomedical Proteomics Laboratory, Institute of Molecular and Cell Biology, Agency for Science, Technology and Research, Singapore, Singapore; [4]Department of Anatomy, Yong Loo Lin School of Medicine, National University of Singapore, Singapore, Singapore; [5]La Trobe University-Proteomics and Metabolomics Platform (LTU-PMP), La Trobe Institute for Molecular Science, La Trobe University, Melbourne, Australia; [6]Epigenes Australia Pty Ltd, Melbourne, Australia; [7]Department of Neuroscience, Johns Hopkins University School of Medicine, Baltimore, United States; [8]Centre for Cardiovascular Biology and Disease Research, La Trobe Institute for Molecular Sciences, La Trobe University, Bundoora, Australia; [9]Baker Heart and Diabetes Institute, Melbourne, Australia; [10]The Paul Feder laboratory for Alzheimer's disease research at the Mina and Everard Goodman Faculty of Life Sciences, and the Leslie and Susan Gonda Multidisciplinary Brain Research Center, Bar Ilan University, Ramat Gan, Israel; [11]Laboratory for Neuroimmunology in Health and Disease, Center for Medical Research on Innovation and Translation, The Second Affiliated Hospital, School of Medicine, South China University of Technology, Guangzhou, China; [12]Department of Microbiology and Immunology, School of Medicine; Institute of Geriatric Immunology, School of Medicine, Jinan University, Guangzhou, China; [13]Department of Pharmacology, Yong Loo Lin School of Medicine, National University of Singapore, Singapore, Singapore; [14]School of Pharmacy, Sungkyunkwan University, Suwon, Republic of Korea

*For correspondence:
jayanthag@imcb.a-star.edu.sg
(JG);
g.arumugam@latrobe.edu.au
(TVA)

†These authors contributed
equally to this work

Reviewing Editor: Rozalyn
M Anderson, University of
Wisconsin–Madison, United
States

## eLife Assessment

This is a **solid** paper on intermittent fasting that will be of interest to readers. The data presented are certainly **valuable** as a resource. The findings of both shared and tissue-specific signatures, both at the proteomic and transcriptomic levels, align well with what has been established and bring new insight into metabolic adaptation and its consequences in muscle, cortex, and liver. The organ specific changes unveiled by proteomics in response to IF reveal unique rewiring of metabolic, signaling and physiological function.

**Abstract** Intermittent fasting (IF) has emerged as a powerful dietary intervention with profound metabolic benefits, yet the tissue-specific molecular mechanisms underlying these effects remain poorly understood. In this study, we employed comprehensive proteomics and transcriptomics analysis to investigate the systemic and organ-specific adaptations to IF in male C57BL/6 mice. Following a 16 hr daily fasting regimen (IF16) over 4 months, IF reduced blood glucose, HbA1c, and cholesterol levels while increasing ketone bodies, indicative of enhanced metabolic flexibility. Proteomic profiling of the liver, skeletal muscle, and cerebral cortex revealed tissue-specific responses, with the liver exhibiting the most pronounced changes, including upregulation of pathways involved in fatty acid oxidation, ketogenesis, and glycan degradation, and downregulation of steroid hormone and cholesterol metabolism. In muscle, IF enhanced pyruvate metabolism, fatty acid biosynthesis, and AMPK signaling, while suppressing oxidative phosphorylation and thermogenesis. The cerebral cortex displayed unique adaptations, with upregulation of autophagy, PPAR signaling, and metabolic pathways, and downregulation of TGF-beta and p53 signaling, suggesting a shift toward energy conservation and stress resilience. Notably, Serpin A1c emerged as the only protein commonly upregulated across all three tissues, highlighting its potential role in systemic adaptation to IF. Integrative transcriptomic and proteomic analyses revealed partial concordance between mRNA and protein expression, underscoring the complexity of post-transcriptional regulation. Shared biological signaling processes were identified across tissues, suggesting unifying mechanisms linking metabolic changes to cellular communication. Our findings reveal both conserved and tissue-specific responses by which IF may optimize energy utilization, enhance metabolic flexibility, and promote cellular resilience.

## Introduction

In recent decades, the global aging population has increased dramatically, resulting in a corresponding rise in age-related diseases. Age-related decline in the functional capacity of vital organs, including the brain, liver, and skeletal muscle, is well documented and attributed to several interrelated mechanisms including cellular senescence, oxidative stress, mitochondrial dysfunction, and diminished regenerative capacity (*López-Otín et al., 2023*). Intermittent fasting (IF) and calorie restriction (CR) have emerged as promising interventions that can improve health and counteract disease processes in rodents, non-human primates, and humans (*Colman et al., 2009*; *Hatori et al., 2012*; *Mattison et al., 2017*; *Sutton et al., 2018*; *Teong et al., 2023*; *Mattson, 2022*). Evidence suggests that IF confers protection against a range of age-associated diseases by bolstering cellular stress resistance, enhancing autophagy, and promoting cell growth and plasticity (*Arumugam et al., 2010*; *Longo and Mattson, 2014*; *Longo and Anderson, 2022*; *Mattson, 2025*). During IF, liver glycogen stores are depleted, leading to low circulating glucose levels, while adipocytes mobilize fatty acids (*Anton et al., 2018*). These fatty acids are subsequently converted in the liver into the ketone bodies β-hydroxybutyrate (BHB) and acetoacetate (AcAc), which enter the bloodstream and are utilized as alternative energy substrates by neurons and other cell types. This metabolic shift, characterized by the transition from carbohydrate and glucose utilization to the reliance on fatty acids and ketones, is referred to as the metabolic switch (*Mattson et al., 2018*).

Intermittent metabolic switching enhances physiological functions across multiple organ systems by regulating the expression of specific genes and proteins (*Kim et al., 2018a*; *Ng et al., 2022*; *Arumugam et al., 2023*; *Mattson, 2025*). IF-induced metabolic switching has been shown to upregulate genes involved in organ remodeling, influencing key metabolic pathways such as AMPK signaling, insulin signaling, and cellular signaling cascades involving protein chaperones and trophic factors (*Mattson et al., 2018*). These IF-mediated alterations in gene expression are likely influenced by changes in the epigenetic landscape, including DNA methylation and histone modifications (*Asif et al., 2020*; *Ng et al., 2022*; *Selvaraji et al., 2022*; *Tabassum et al., 2025*). Furthermore, our recent deep proteomic and phosphoproteomic analyses of heart tissue revealed multiple adaptive responses of cardiac cells to IF (*Arumugam et al., 2023*). While these studies shed light on the mechanistic foundations of IF, a comprehensive and systematic analysis is needed to identify the unique and shared protein changes of cells in various organs in response to IF.

This study explores the proteomic and transcriptomic responses to IF of three distinct organ systems: the liver, the brain, and skeletal muscle. By systematically analyzing expression patterns in

response to IF, we identified differentially expressed proteins (DEPs) and investigated their roles in metabolic and cell signaling pathways. Through comparative analysis of these proteins across the three organs, we uncovered critical processes that may underlie the health benefits of IF. Our integrative approach revealed both common and tissue-specific protein network of common and unique proteins that link IF to systemic physiological effects. To our knowledge, this is the first study that provides a comprehensive understanding of how IF modulates proteomes across multiple organs, offering novel insights into the molecular mechanisms driving its health-promoting effects.

## Results and discussion

Male C57BL/6 mice were fed a normal chow diet and randomly assigned to AL or daily IF16 schedules beginning at 6 weeks of age. The study design, including the timing of experimental interventions and blood and tissue collections, is summarized in *Figure 1—figure supplement 1A*. To determine the extent to which 8 hr IF affects energy metabolism, we measured the blood glucose, HbA1c, ketone and cholesterol levels, and body weight of all mice during the 4-month dietary intervention period. Mice in the IF group exhibited a significant decrease in blood glucose and HbA1c levels compared to the AL group (*Figure 1—figure supplement 1B and C*). IF mice exhibited blood ketone levels that were two- to threefold greater than mice in the AL group (*Figure 1—figure supplement 1D*). In addition, IF mice exhibited lower blood cholesterol levels compared to AL (*Figure 1—figure supplement 1E*). IF animals exhibited significantly lower body weight and body weight changes compared to AL mice during the 4-month dietary intervention period (*Figure 1—figure supplement 1F and G*).

### Proteomic profiling

To gain a comprehensive understanding of the mechanisms affected by IF in the liver, muscle, and brain, we performed proteome profiling of these tissues using seven biological replicates from each dietary group (AL = 7, IF = 7; *Figure 1*, *Figure 1—figure supplement 2A–D*). Mass spectrometry-based quantitative proteomics analysis of these tissues identified a total of 4777 proteins in the liver (*Supplementary file 1*). Of these, 835 proteins (17.5%) were differentially expressed in IF liver compared to the control AL group (*Figure 1A and B* and *Supplementary file 1*). In muscle, we identified 2709 proteins, with 155 (5.7%) showing differential expression in IF muscle (*Figure 1C and D* and *Supplementary file 1*). In the cerebral cortex, we identified 6332 proteins, of which 294 (6%) were differentially expressed in IF cerebral cortex (*Figure 1E and F* and *Supplementary file 1*). We next analyzed the DEPs across all organs to assess their biological significance.

In the liver, IF led to the upregulation of 446 proteins, predominantly associated with metabolic pathways. These pathways exhibited a >10-fold enrichment in key processes such as the citrate cycle, glycan degradation, butanoate metabolism, glycine, serine, and threonine metabolism, and fatty acid metabolism, among others (*Figure 1G*). Enhancement of these pathways in response to IF may optimize energy metabolism: the citrate cycle is activated to efficiently oxidize fatty acids and generate ATP, while glycan degradation facilitates the conversion of stored glycans into readily usable sugars. Additionally, increased butanoate metabolism enables the liver to utilize short-chain fatty acids, enhancing its adaptation to fasting conditions (*Zhou et al., 2017*). Interestingly, proteins involved in thermogenesis were also upregulated. Conversely, IF resulted in the downregulation of 389 proteins in the liver, with enrichment observed in pathways such as steroid synthesis and linoleic acid metabolism, alongside other biosynthetic processes (*Figure 1H*). This likely reflects the liver's prioritization of energy-producing pathways over steroid hormone synthesis during fasting (*Seoane-Collazo et al., 2018*). Metabolic switching and altered signaling pathways under IF conditions may suppress enzymes involved in steroidogenesis (*Grasfeder et al., 2009*). This metabolic shift underscores the liver's ability to maintain energy balance and metabolic flexibility in response to limited nutrient availability, highlighting the adaptive mechanisms driven by IF (*Smith et al., 2018*).

Among the 69 proteins upregulated in skeletal muscle in response to IF, pathways such as pyruvate metabolism, fatty acid biosynthesis, and carbon and fatty acid metabolism exhibited over 20-fold enrichment in IF-fed animals compared to AL-fed controls (*Figure 1I*). This upregulation suggests that IF enhances the muscle's capacity for energy production and substrate utilization, particularly through the activation of metabolic pathways critical for processing pyruvate and synthesizing fatty acids. Additionally, key bioenergetic pathways, including AMPK, PPAR, insulin, and glucagon signaling

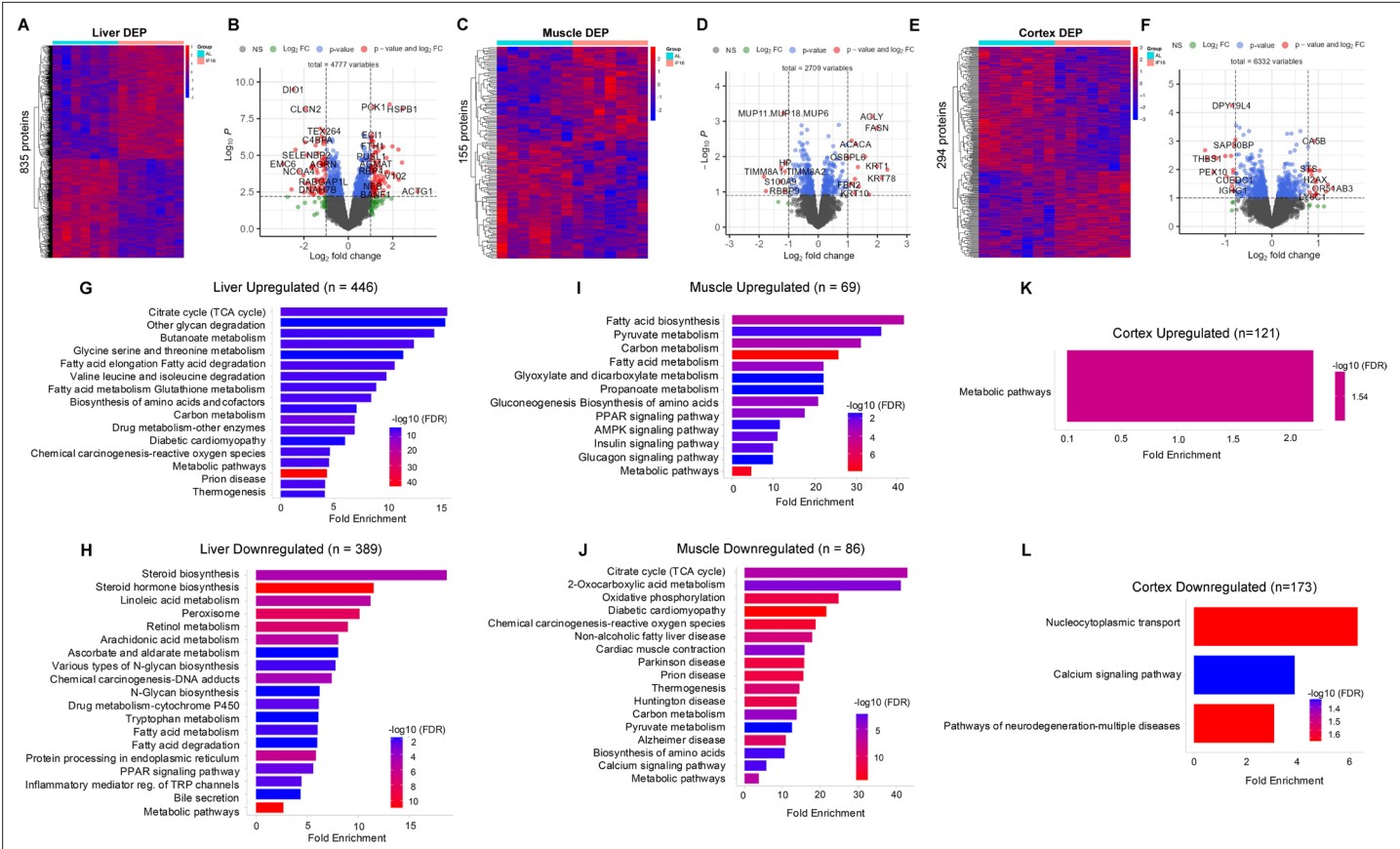

**Figure 1.** Differential protein expression and functional analysis in response to intermittent fasting across multiple tissues. (**A**) Heatmap of proteins significantly differentially expressed in IF group and AL group in liver (**A**), in muscle (**C**), and in cerebral cortex (**E**). The color scale denotes z-score normalized protein abundances. Red denotes higher protein abundance, and blue lower protein abundance. (**B**) Volcan plot of all protein expression in IF groups in comparison to the AL group in liver (**B**), in muscle (**D**), and in cerebral cortex (**F**). Each dot represents a protein. Red dots showing protein with p<0.05 and |log2FC|>1. Blue dots showing proteins with p<0.05 and 0<|log2FC|<1. Green dots showing proteins with p>0.05 and |log2FC >1|. (**G**) Enrichment analysis of modulated processes and pathways in IF groups in comparison to the AL group in both upregulated and downregulated pathways in liver (**G,H**), muscle (**I, J**), and cerebral cortex (**K, L**). The FDR was controlled at 0.05. Each bar represents a functional process or pathway. Red denotes higher protein abundance, and blue lower protein abundance. n=7 mice in each group.

The online version of this article includes the following figure supplement(s) for figure 1:

**Figure supplement 1.** Experimental design and metabolic analysis.

**Figure supplement 2.** 3D principal component analysis (PCA) of proteomic profiles.

were upregulated, perhaps indicating that IF promotes metabolic efficiency in muscle by enhancing energy sensing, lipid metabolism, and glucose homeostasis. Conversely, IF led to the downregulation of 86 proteins in skeletal muscle, which were primarily associated with pathways such as the citrate cycle, oxidative phosphorylation, and thermogenesis (*Figure 1J*). Critically, this coordinated downregulation should not be interpreted as a simple reduction in mitochondrial oxidative capacity. Rather, it likely reflects a profound metabolic adaptation to intermittent fasting, where muscle enters a more efficient bioenergetic state. During the fasting window, the primary fuel shifts from glucose to lipids, requiring efficient fatty acid β-oxidation. The observed downregulation may indicate a remodeling of the mitochondrial proteome to optimize electron flow, thereby reducing proton leak and unnecessary thermogenesis to conserve energy. This reprioritization from a glycolytic to a lipid-oxidative phenotype represents a key adaptation to periodic nutrient scarcity. Notably, muscle's response to IF differs significantly from that of the liver, highlighting tissue-specific adaptations to fasting and feeding cycles. While the liver predominantly upregulates pathways involved in gluconeogenesis, glycan degradation, and fatty acid oxidation, muscle appears to fine-tune its metabolic machinery to balance energy production and substrate utilization in a distinct manner. These findings underscore

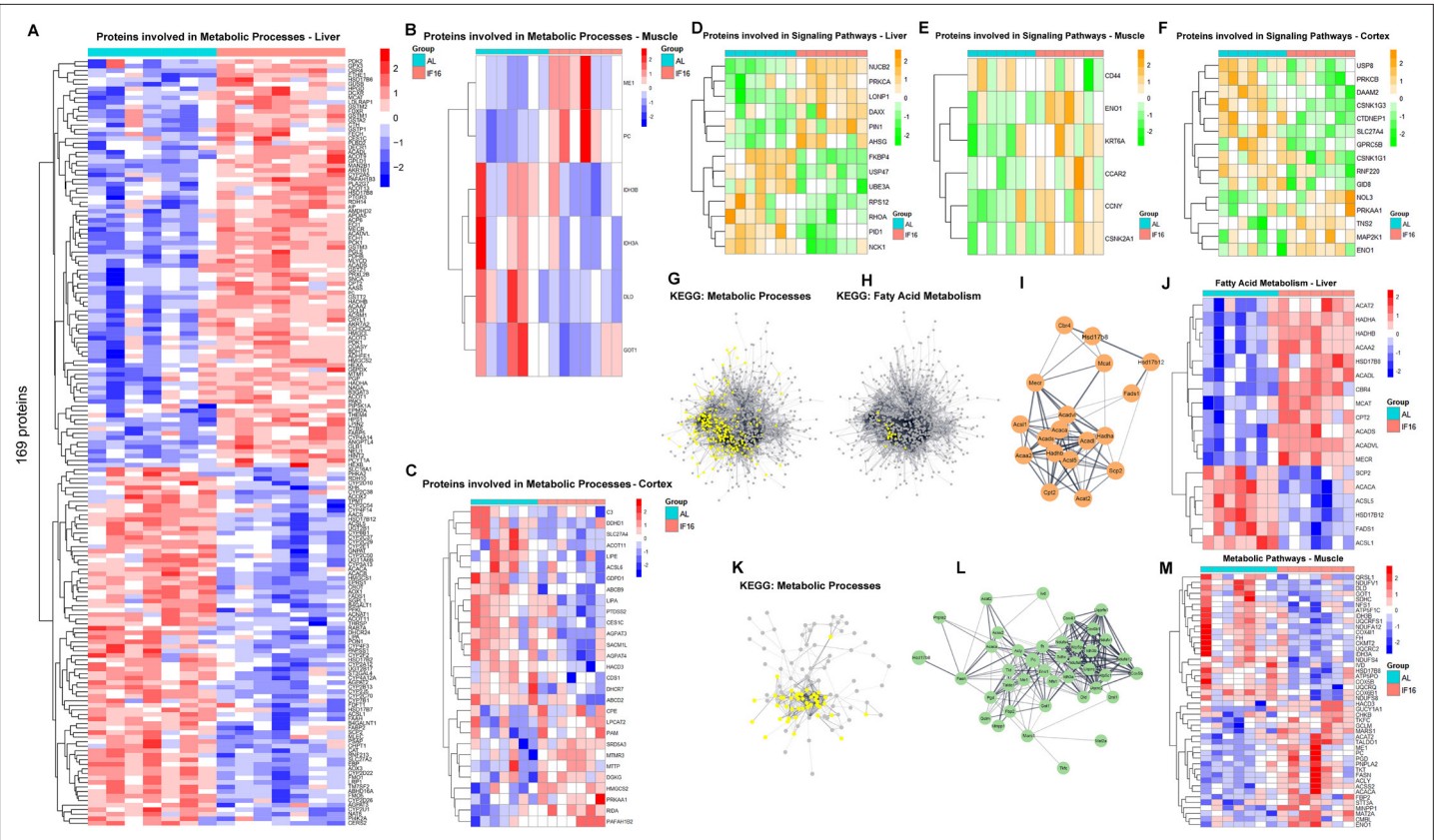

**Figure 2.** Differential expression patterns of proteins, functional analysis, and associated protein networks across multiple tissues in response to intermittent fasting across multiple tissues. (**A**) Heatmap of significantly different proteins involved in metabolic process in IF group in comparison to AL group in liver (**A**), muscle (**B**), and cerebral cortex (**C**). (**D**) Heatmap of significantly different proteins involved in signaling pathways in IF group in comparison to AL group in liver (**D**), muscle (**E**), and cerebral cortex (**F**). (**G**) Functional transition network that shows metabolic processes (**G**) and fatty acid metabolism (**H, I**) in response to IF in liver and metabolic processes (**K, L**) in response to IF in muscle. (**J**) Heatmap of fatty acid metabolism-related protein expression in response to IF in liver. (**M**) Heatmap of metabolic pathways-related protein expression in response to IF in muscle. The FDR was controlled at 0.05 for GO biological processes and pathway databases. The functional transition network was based on gene set similarity as p-value <0.01. n=7 mice in each group.

The online version of this article includes the following figure supplement(s) for figure 2:

**Figure supplement 1.** Network mapping of IF-responsive proteins in liver.

**Figure supplement 2.** Network mapping of IF-responsive proteins in muscle Protein-protein interaction networks generated using STRING analysis highlight multiple functional pathways associated with proteins differentially expressed in muscle in response to intermittent fasting.

**Figure supplement 3.** Network mapping of IF-responsive proteins in cortex Protein-protein interaction networks generated using STRING analysis highlight multiple functional pathways associated with proteins differentially expressed in cortex in response to intermittent fasting.

the organ-specific effects of IF, revealing how different tissues coordinate their metabolic responses to optimize energy balance and physiological function under restricted feeding conditions.

Unlike the liver and muscle, 121 upregulated proteins in the cerebral cortex were primarily linked to metabolic pathways (*Figure 1K*). In contrast, 173 proteins were downregulated in the cerebral cortex, with significant associations observed in pathways such as nucleocytoplasmic transport, calcium signaling, and processes related to neurodegeneration (*Figure 1L*). These findings highlight the distinct metabolic and functional adaptations of the cortex to IF, which differ markedly from those observed in the liver and muscle.

## Identification of altered proteins and metabolic pathway dynamics

To investigate the major metabolic and signaling pathways responsive to IF in the liver, cerebral cortex, and muscle, we further analyzed DEPs in these tissues. Notably, the liver exhibited the highest number of DEPs (*Figure 2A*, *Supplementary file 1* and *Supplementary file 2*), which are primarily involved in

metabolic processes. In contrast, fewer changes were observed in muscle (*Figure 2B*, *Supplementary file 2Supplementary file 1* and *Supplementary file 2*) and the cerebral cortex (*Figure 2C*, *Supplementary file 1* and *Supplementary file 2*). In response to IF, the liver exhibits upregulation of several proteins that are intricately involved in key metabolic pathways (*Figure 2A*). Proteins such as PDK2, PDHB, and MECR play critical roles in mitochondrial energy metabolism, including pyruvate metabolism and fatty acid oxidation, which are essential for maintaining energy homeostasis during fasting (*Luo et al., 2022a*; *Jiang et al., 2023*; *Qian et al., 2020*). Enzymes like ACADL, DECR1, and BDH1 are directly involved in lipid metabolism, facilitating the breakdown and utilization of fatty acids as an energy source (*Diekman et al., 2021*; *Mäkelä et al., 2019*; *Williams et al., 2024*). Additionally, CTH, CES1C, and GSTMs (e.g. GSTM2, GSTM3) contribute to detoxification and antioxidant pathways, protecting the liver from oxidative stress induced by metabolic shifts (*Katsouda et al., 2023*; *Gan et al., 2023*; *Fafián-Labora et al., 2020*). Proteins such as GPX3, FDXR, and AKR1B1 further support redox balance and stress response mechanisms (*Wu et al., 2023*; *Mohibi et al., 2024*; *Yu et al., 2024*). Meanwhile, FABP5, PCYT1A, and LPIN2 are associated with lipid transport and phospholipid biosynthesis, ensuring proper lipid handling and membrane integrity (*Seo et al., 2020*; *Haider et al., 2018*; *Zhang et al., 2019*). Finally, HEXA, CTBS, and GUSB are linked to glycosphingolipid and carbohydrate metabolism, highlighting the liver's role in managing glucose and glycoconjugate processing during fasting (*Montgomery et al., 2023*; *Persichetti et al., 2012*; *Bramwell et al., 2015*). In response to IF, the downregulation of key proteins in the liver reflects a shift in metabolic priorities and reduced activity in specific pathways. For instance, CYP450 enzymes such as CYP2E1, CYP3A13, and CYP7B1 are involved in xenobiotic metabolism and bile acid synthesis, suggesting a potential reduction in detoxification processes during fasting (*Esteves et al., 2021*). Similarly, ACACA and ACACB, key enzymes in fatty acid synthesis, are downregulated, aligning with the suppression of lipogenesis when energy intake is limited (*Dong et al., 2024*; *Bhattacharjee et al., 2020*). Proteins like HMGCS1 and FDFT1, which are critical for cholesterol biosynthesis, also show reduced expression, indicating a slowdown in sterol production (*Qin et al., 2024*; *Ha and Lee, 2020*). Additionally, PFKL, a regulator of glycolysis, is downregulated, reflecting a decreased reliance on glucose metabolism during fasting (*Meng et al., 2024*). The downregulation of FADS1 and ACSL1, involved in fatty acid desaturation and activation, further highlights a shift away from lipid processing and storage (*Reynolds et al., 2020*; *Wang et al., 2024*).

The downregulation of key proteins in muscle tissue reflects a shift in metabolic activity and energy utilization (*Figure 2B*). For instance, DLD and IDH3A/IDH3B are crucial components of the TCA cycle, suggesting a reduction in mitochondrial oxidative metabolism during fasting (*Capitanio et al., 2017*; *Liu et al., 2020a*). Similarly, GOT1, which plays a role in amino acid metabolism and the malate-aspartate shuttle, is downregulated, indicating a potential decrease in amino acid catabolism and nitrogen handling (*Sun et al., 2019*).

The upregulation of key proteins in the cerebral cortex highlights adaptive changes in metabolic and signaling pathways (*Figure 2C*). PRKAA1, a subunit of AMPK, is upregulated, indicating enhanced cellular energy sensing and regulation, which promotes energy conservation and catabolic processes (*Yang et al., 2020*). HMGCS2, a critical enzyme in ketogenesis, suggests an increased production of ketone bodies, providing an alternative energy source for the brain during fasting (*Asif et al., 2022*). LPCAT2 and PAFAH1B2, involved in phospholipid metabolism and inflammatory signaling, respectively, may play roles in maintaining membrane integrity and modulating neuroinflammatory responses (*Cotte et al., 2018*; *Sudarov et al., 2018*). Additionally, PAM and CPE are associated with neuropeptide processing, potentially influencing neuronal signaling and stress adaptation (*Powers et al., 2021*; *Chen et al., 2023*). In response to IF, the downregulation of key proteins in the cortex reflects a shift in metabolic and signaling pathways to adapt to reduced nutrient availability. ACSL6 and SLC27A4, involved in fatty acid activation and transport, respectively, show reduced expression, suggesting a decrease in lipid metabolism and utilization in the cortex during fasting (*Fernandez et al., 2021*; *Anderson and Stahl, 2013*). LIPE and LIPA, which play roles in lipid hydrolysis and cholesterol metabolism, are also downregulated, indicating a suppression of lipolytic and sterol-related processes (*Adom et al., 2024*; *Arnaboldi et al., 2020*). CES1C, an enzyme involved in ester hydrolysis, further supports the downregulation of lipid metabolism pathways (*Gan et al., 2023*). Additionally, PTDSS2, associated with phospholipid synthesis, is reduced, pointing to a slowdown in membrane lipid production (*Yang et al., 2019*).

The liver undergoes significant changes in signaling pathways mediated by the upregulation and downregulation of key proteins (*Figure 2D*, *Supplementary file 1* and *Supplementary file 2*) in response to IF. Upregulated proteins such as NUCB2 and PRKCA play roles in calcium signaling and cellular stress responses, potentially enhancing adaptive mechanisms to maintain energy homeostasis (*Liu et al., 2020b*; *Wang et al., 2017*). LONP1, a mitochondrial protease, supports protein quality control and metabolic efficiency, while DAXX and PIN1 are involved in apoptosis regulation and cell cycle signaling, respectively, suggesting a balance between cell survival and stress adaptation (*Zhao et al., 2022*; *Tang et al., 2020*; *Chen et al., 2018a*). On the other hand, downregulated proteins like FKBP4 and USP47, which are associated with steroid receptor signaling and protein stabilization, indicate a reduction in steroid hormone activity and protein turnover (*Kageyama et al., 2021*; *Lei et al., 2021*). UBE3A and RPS12, linked to ubiquitin-mediated degradation and ribosomal function, reflect a slowdown in protein synthesis and degradation pathways (*Avagliano Trezza et al., 2021*; *Kale et al., 2018*). Additionally, the downregulation of RHOA, PID1, and NCK1, which regulate cytoskeletal dynamics and cell signaling, points to reduced cellular motility and signaling activity (*Li et al., 2012*; *Yang et al., 2023a*; *Li et al., 2014*).

The upregulation and downregulation of specific proteins in muscle tissue reflect adaptive changes in signaling pathways to optimize energy utilization and cellular function in response to IF (*Figure 2E*, *Supplementary file 1* and *Supplementary file 2*). Upregulated proteins such as CCNY and CSNK2A1 are involved in cell cycle regulation and phosphorylation-dependent signaling, suggesting enhanced cellular repair and stress adaptation mechanisms (*Opacka et al., 2023*; *Yang et al., 2023b*). ENO1, a key glycolytic enzyme, indicates a potential shift in energy metabolism to maintain ATP production under nutrient scarcity (*Huppertz et al., 2022*). CCAR2 and KRT6A are associated with DNA damage response and cytoskeletal integrity, respectively, highlighting the muscle's efforts to maintain structural stability and genomic fidelity during fasting (*Iyer et al., 2022*; *Rorke et al., 2015*). Conversely, the downregulation of CD44, a cell surface glycoprotein involved in cell adhesion and signaling, suggests reduced inflammatory signaling and cellular interactions, which may help conserve energy and minimize unnecessary cellular activity (*Wolf et al., 2020*).

The cerebral cortex exhibits adaptive changes in signaling pathways mediated by the upregulation and downregulation of key proteins, many of which contribute to beneficial neuroprotective and metabolic effects (*Figure 2F*, *Supplementary file 1* and *Supplementary file 2*). Upregulated proteins such as PRKAA1 enhance energy sensing and promote cellular resilience by activating pathways that support mitochondrial function and energy efficiency (*Yang et al., 2018*). MAP2K1 supports cell survival and growth signaling (*Mizuno et al., 2023*). NOL3 and TNS2 are involved in anti-apoptotic signaling and cytoskeletal organization, respectively, further promoting neuronal stability and repair (*Wu et al., 2024*; *Cheng et al., 2018*). Conversely, the downregulation of proteins such as PRKCB and USP8 reduces inflammatory and stress-related signaling, while SLC27A4 and GPRC5B indicate a shift away from lipid metabolism and non-essential signaling (*Griss et al., 2019*; *Zhao et al., 2020*; *Maekawa et al., 2015*; *Kim et al., 2018b*).

To better understand how metabolic protein networks are restructured in response to IF, we compiled protein-protein interactions (PPIs) for all IF-responsive proteins and investigated their functional interplay in the liver (*Figure 2G–J*) and muscle (*Figure 2K–M*). Our analysis identified closely associated protein networks that are associated with fatty acid metabolism in the liver and analyzed their expression levels in response to IF (*Figure 2I and J*). IF induces significant metabolic adaptations in muscle tissue, as revealed by the upregulation and downregulation of key proteins identified through PPI/KEGG networks. Upregulated proteins such as PGD, TKT, FASN, ACLY, ACACA, MAT2A, ENO1, and GCLM highlight a shift toward enhanced lipid metabolism, redox balance, and stress protection (*Wakai et al., 2017*; *Tian et al., 2020*; *Schroeder et al., 2021*; *Lin et al., 2013*; *Luo et al., 2022b*; *Krejsa et al., 2010*). PGD and TKT support NADPH production, while FASN, ACLY, and ACACA drive fatty acid synthesis, reflecting a response to energy fluctuations (*Currie et al., 2013*; *TeSlaa et al., 2023*). MAT2A and GCLM indicate increased methionine metabolism and glutathione synthesis, safeguarding against oxidative stress, while ENO1 underscores the persistence of glucose metabolism (*Cheng et al., 2024*). Conversely, downregulated proteins like IDH3B, FH, DLD, NDUFV1, COX4I1, IVD, and CKMT2, involved in the TCA cycle and mitochondrial energy production, suggest a reduced reliance on oxidative metabolism as muscle tissue conserves energy and prioritizes alternative fuel sources (*Zhu et al., 2022*; *Solaimuthu et al., 2022*; *Fuhrmann and Brüne, 2017*;

*Čunátová et al., 2021*; *Lin et al., 2024*). Together, these changes illustrate a coordinated metabolic shift, balancing energy production, lipid utilization, and stress responses to maintain homeostasis during fasting. This dynamic adaptation underscores the metabolic flexibility of muscle tissue and provides valuable insights into the molecular mechanisms underlying IF.

Furthermore, we have identified additional metabolic and bioenergetic networks modulated by IF across different tissues. In the liver (*Figure 2—figure supplement 1*), these networks include FGF and insulin signaling, glutamate metabolism, and nucleoside diphosphate metabolic processes, highlighting the liver's central role in energy regulation and metabolic homeostasis. In muscle (*Figure 2—figure supplement 2*), processes such as biosynthesis and protein glycosylation were prominent, reflecting tissue-specific adaptations to nutrient availability. In the cortex (*Figure 2—figure supplement 3*), pathways related to brain development, neurotransmitter transport, synaptic vesicle maturation, and cytoskeleton organization were observed, underscoring the brain's adaptive responses to fasting. Collectively, these findings demonstrate the systemic impact of IF, revealing tissue-specific metabolic reprogramming that supports energy conservation, cellular maintenance, and functional adaptation.

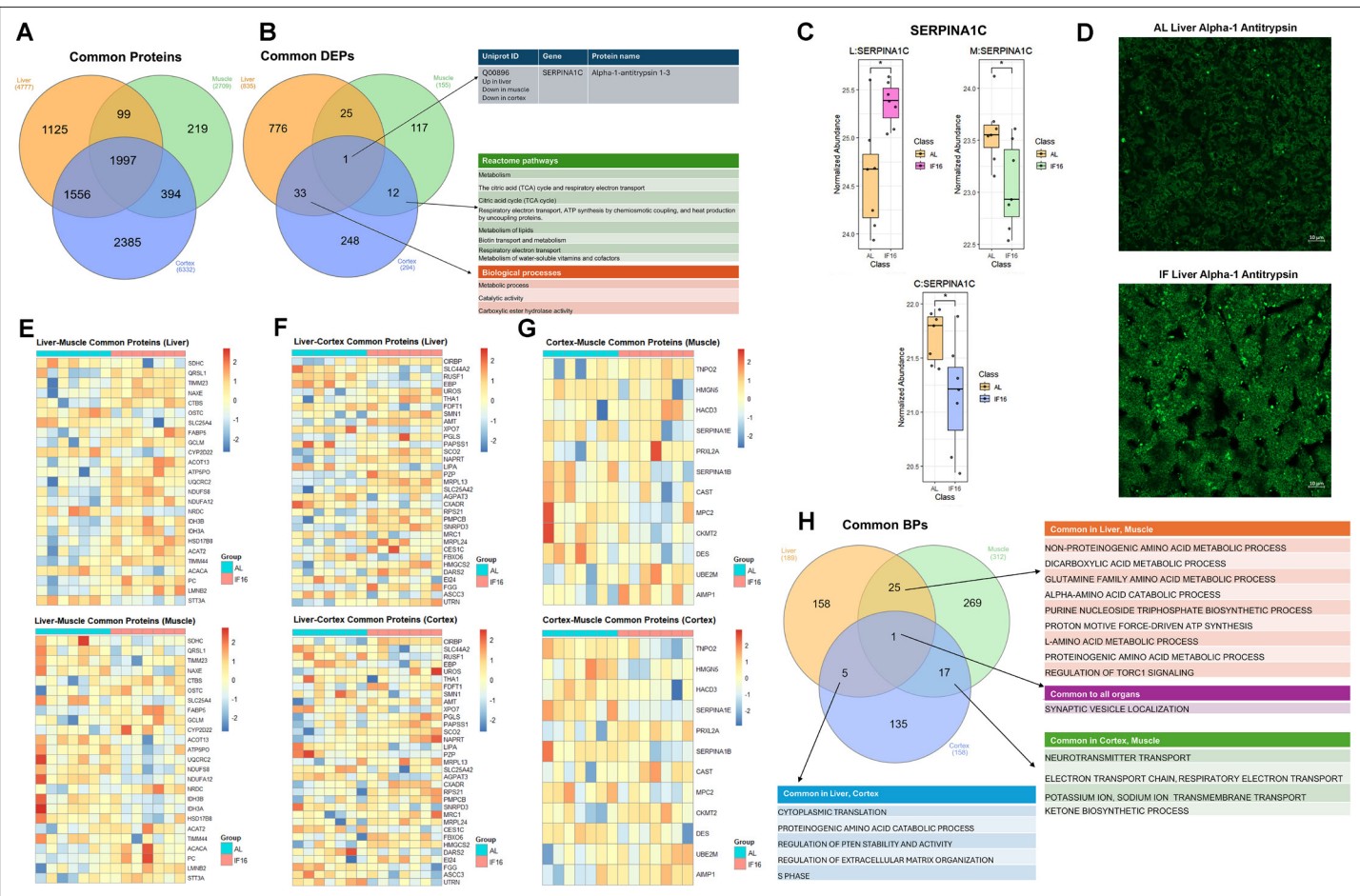

**Figure 3.** Integrative analysis of protein expression and functional enrichment across multiple tissues. (**A**) Venn diagram illustrating common proteins shared among the liver, muscle, and cerebral cortex. (**B**) Venn diagram depicting common differentially expressed proteins across the liver, muscle, and cortex. (**C**) Expression levels of Serpin A1c in the liver, muscle, and cerebral cortex. (**D**) Alpha1-antitrypsin staining in green in the liver of both AL and IF mice. n=3 mice in each group. (**E**) Heatmap of overlapping differentially expressed proteins between the liver and muscle. (**F**) Heatmap of overlapping differentially expressed proteins between the liver and cerebral cortex. (**G**) Heatmap of overlapping differentially expressed proteins between the cerebral cortex and muscle. (**H**) Venn diagram highlighting common biological processes shared among the liver, muscle, and cerebral cortex. Data was presented as mean ± SEM with p<0.05 was defined as significantly different. n=7 mice in each group.

The online version of this article includes the following figure supplement(s) for figure 3:

**Figure supplement 1.** Shared differentially expressed proteins across organs in response to intermittent fasting.

## Identification of shared and tissue-specific biological processes

While the health benefits of IF are well documented, the underlying molecular mechanisms driving these effects remain poorly understood, particularly whether they are mediated through shared or tissue-specific pathways. To address this knowledge gap, we conducted a comparative analysis to identify unique and shared biological processes across the liver, muscle, and cortex in response to IF. First, we compared all proteins expressed in the IF groups across the liver, muscle, and cerebral cortex, identifying 1997 proteins commonly expressed in response to IF (*Figure 3A*, *Supplementary file 3*). Next, to elucidate the shared and tissue-specific mechanisms of IF, we focused on DEPs that showed significant changes compared to the AL control group (*Figure 3B*, *Supplementary file 3*). Our analysis revealed that Serpin A1c was the only protein commonly dysregulated across all three organs, suggesting a potentially central role for this protein in the systemic response to IF. The observation that Serpin A1c levels increase in the liver but decrease in muscle and cerebral cortex in response to IF suggests a tissue-specific regulatory role for this protein during metabolic adaptation to IF (*Figure 3C*, *Supplementary file 3*). In the liver, the upregulation of Serpin A1c, a serine protease inhibitor, may reflect a protective mechanism to modulate inflammation or protein degradation as the body shifts toward fat metabolism and ketogenesis during fasting (*Okagawa et al., 2024*). Conversely, its reduction in muscle and cerebral cortex could indicate a reduced need for protease inhibition in these tissues, possibly to facilitate protein breakdown for energy or to utilize resources under fasting conditions (*Sanfeliu et al., 2019*; *Pattamaprapanont et al., 2024*). These differential responses highlight how IF can trigger distinct molecular adjustments across tissues, reflecting their unique metabolic demands and contributions to whole-body homeostasis. To validate this finding, we performed immunohistochemistry to assess Serpin A1c expression in liver tissues obtained from both IF and AL fed animals. The results confirmed that IF upregulates the expression levels of Serpin A1c in the liver (*Figure 3D*).

Next, we conducted an organ-to-organ comparison to identify common DEPs in response to IF. Between the liver and muscle, we identified 25 proteins that were differentially expressed (*Figure 3E*, *Supplementary file 3*). Notably, LMNB2 (involved in nuclear envelope integrity and cell cycle regulation; *Ji et al., 2022*), PC (a key enzyme in gluconeogenesis and anaplerosis; *Hughey and Crawford, 2019*), and CTBS (associated with lysosomal function and glycoprotein metabolism; *Yadati et al., 2020*) were upregulated in both organs, suggesting their roles in supporting metabolic adaptation and cellular homeostasis during fasting. In contrast, SDHC (a subunit of mitochondrial complex II, critical for the electron transport chain; *Bandara et al., 2021*) was downregulated in both tissues, potentially reflecting a reduction in oxidative phosphorylation to conserve energy under nutrient scarcity. The remaining common DEPs exhibited opposing expression patterns, highlighting distinct, organ-specific responses to IF.

Between the liver and cerebral cortex, we identified 33 proteins that were differentially expressed in response to IF (*Figure 3F*, *Supplementary file 3*). Among the upregulated proteins, CIRBP (involved in stress response and RNA stabilization; *Zhu et al., 2024*), UROD (essential for heme biosynthesis; *Nilsson et al., 2009*), SCO2 (critical for mitochondrial function and energy production; *Sung et al., 2010*), NAPRT (a key enzyme in NAD +biosynthesis; *Carreira et al., 2023*), MRPL13 (involved in mitochondrial protein synthesis; *Cámara et al., 2011*), RPS21 (essential for protein translation; *Dinh et al., 2021*), PMPCB (crucial for mitochondrial protein import; *Michaelis et al., 2022*), and HMGCS2 (a rate-limiting enzyme in ketogenesis; *Kim et al., 2019a*) were upregulated in both organs. Upregulation of these proteins collectively suggests enhanced stress resilience, mitochondrial function, and metabolic adaptation during fasting. Among the downregulated proteins, RUSF1 (implicated in cell signaling; *Huttlin et al., 2010*), LIPA (involved in lipid metabolism; *Lettieri Barbato et al., 2013*), XPO7 (associated with nuclear transport; *Aksu et al., 2018*), and AGPAT3 (involved in phospholipid biosynthesis; *Zhou et al., 2024*) were reduced, potentially reflecting a shift in energy utilization and cellular homeostasis. These findings highlight the coordinated yet distinct responses of the liver and cerebral cortex to IF, emphasizing the interplay between shared and tissue-specific mechanisms in metabolic reprogramming. The remaining shared DEPs displayed divergent expression profiles, underscoring the tissue-specific adaptations that occur in response to IF.

Among the 12 DEPs identified between the cortex and muscle in response to IF (*Figure 3G*, *Supplementary file 3*), PRXL2A (involved in oxidative stress response; *Chen et al., 2019*), UBE2M (critical for protein ubiquitination and degradation; *Zhou et al., 2018*), and AIMP1 (associated with

inflammation and angiogenesis; *Zhou et al., 2020*) were upregulated. Upregulation of these proteins suggests enhanced cellular stress resilience, protein homeostasis, and regulatory signaling in both tissues during fasting. Conversely, the downregulated proteins included DES (essential for muscle structural integrity; *Agnetti et al., 2022*), MPC2 (crucial for pyruvate transport and energy metabolism; *Pujol et al., 2023*), Serpin A1b (involved in protease inhibition and inflammation; *Bidooki et al., 2023*), and Serpin A1e (also implicated in protease regulation; *Zhang et al., 2022*). The reduction of these proteins may reflect tissue-specific adaptations, such as altered structural maintenance in muscle and modulated inflammatory responses in the cortex. These findings highlight the distinct yet coordinated molecular mechanisms underlying tissue-specific responses to IF.

We also compared the expression levels of all common DEPs in response to IF relative to AL feeding across the liver, muscle, and cerebral cortex to identify the organs in which these proteins were most highly regulated (*Figure 3—figure supplement 1*). Additionally, we analyzed the common biological processes modulated by IF across all three tissues. Intriguingly, synaptic vesicle localization was the only biological process shared among the liver, muscle, and cortex (*Figure 3H*, *Supplementary file 3*). This finding suggests a potential role for synaptic vesicle regulation in the systemic adaptation to IF, possibly linking metabolic changes to neuronal regulation of peripheral organs. Although the liver and muscle are not traditionally associated with synaptic function, this conserved process may indicate a broader, unifying neural mechanisms underlying organ responses to nutrient stress.

## Analysis of RNA-protein concordance

Having established that IF modulates the proteome in the liver, muscle, and cortex, we next investigated the transcriptome to explore IF-induced effects in these organs. To this end, we performed RNA-seq-based transcriptome profiling across all tissues and compared the differentially expressed transcripts with their corresponding proteomic profiles (*Figure 4—figure supplements 1 and 2*). Proteomic and transcriptomic integration reveals that IF induces a co-regulated program of transcriptional and proteomic upregulation in the liver, indicative of a profound metabolic and homeostatic shift. This activated network encompasses a broad spectrum of cellular functions, including enhanced lysosomal catabolism, ubiquitin signaling, and mitochondrial biogenesis and function. Key metabolic pathways are prominently engaged, as seen in the upregulation of gluconeogenesis, lipid droplet regulation, fatty acid activation, and lipoprotein metabolism. Concurrent increases in the NAD +salvage enzyme, the endoplasmic reticulum electron donor for biosynthesis and detoxification, the drug efflux transporter, and the iron storage protein, collectively point to a liver state that is primed for energy production, cellular repair, detoxification, and adaptive stress resistance under IF (*Figure 4A and B*, *Figure 4—figure supplement 3A*). Transcriptional and proteomic downregulation in the liver in response to IF highlights a concerted retreat from specific biosynthetic and metabolic processes. The suppressed signatures indicate a marked reduction in cholesterol synthesis, evidenced by the rate-limiting enzyme Hmgcs1, and a downscaling of thyroid hormone activation via Dio1. This is accompanied by decreased expression of proteins involved in vascular and epithelial barrier function and complement regulation. Furthermore, IF suppresses genes tied to nutrient processing, including galactose metabolism and aminoacylation for protein translation. Collectively, this downregulated network reflects an IF-induced shift away from anabolic pathways, structural maintenance, and specific metabolic loops, allowing the liver to prioritize core energy-mobilizing functions (*Figure 4A and B*, *Figure 4—figure supplement 3A*).

Proteomic and transcriptomic integration in skeletal muscle delineates a comprehensive, biphasic transcriptional and proteomic remodeling program in response to IF. The downregulated network orchestrates a metabolic and functional shift away from anabolism and excitability, characterized by suppression of potassium channel subunit Kcnj11, mitotic kinase Aurka, structural proteins Ank1 and Flnb, and the mitochondrial complex I subunit Ndufs4. This is accompanied by downregulation of key regulators, including the circadian repressor Nr1d1 (*Boulinguiez et al., 2022*) and the signaling adapters Shc2 and Grip2 (*Jiang et al., 2003*; *Fouillade et al., 2013*), reinforcing a state of reduced biosynthetic and electrical activity (*Figure 4C and D*). Conversely, a robustly upregulated program signals a pronounced metabolic re-prioritization. This includes a strong induction of pyruvate dehydrogenase kinase 4 (Pdk4), promoting a shift from glucose to fatty acid oxidation, and the nuclear receptor Nr4a1, a key mediator of lipid metabolism and mitochondrial biogenesis (*Kanzleiter et al., 2010*). Upregulation of genes involved in protein maturation, ubiquitin processing, methionine

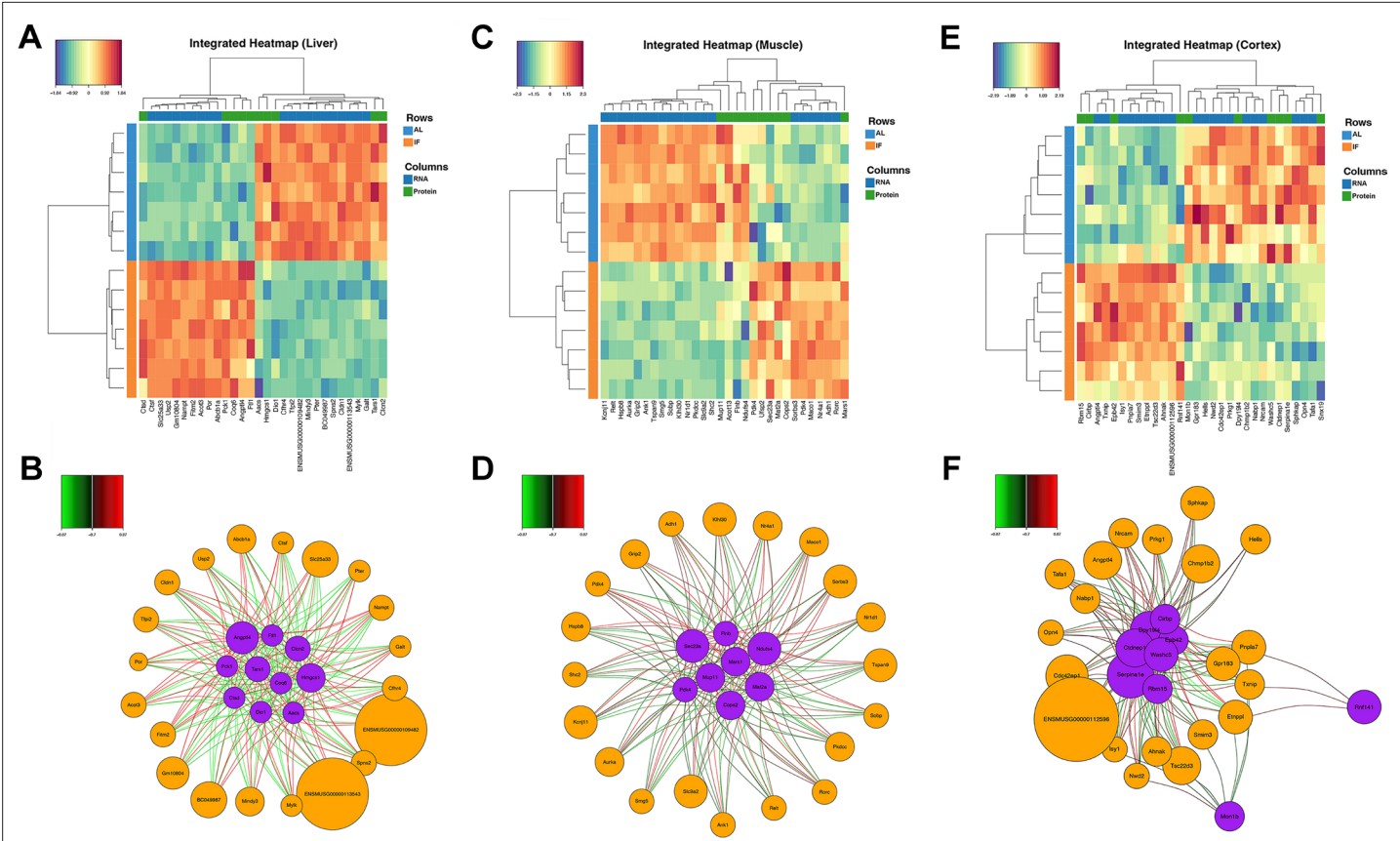

**Figure 4.** Multi-omics integration reveals co-regulated transcriptomic and proteomic networks in response to intermittent fasting across multiple tissues. Integrated heatmaps displaying the expression profiles of the robust molecular signatures selected by DIABLO in liver (**A**), muscle (**C**), and cerebral cortex (**E**). Relevance interaction networks visualizing the functional connectivity and high-confidence correlations between transcripts and proteins in liver (**B**), muscle (**D**), and cerebral cortex (**F**). The integrated signatures were identified using supervised N-integration block sparse PLS-DA (DIABLO) with 10-repeated sevenfold cross-validation to minimize the balanced error rate. n=7 mice in each group.

The online version of this article includes the following figure supplement(s) for figure 4:

**Figure supplement 1.** 3D principal component analysis (PCA) of transcriptomic profiles.

**Figure supplement 2.** Correlation between transcriptomic and proteomic profiles in the integrated DIABLO model.

**Figure supplement 3.** Gene ontology (GO) enrichment analysis of the integrated multi-omics signatures.

metabolism, and cytoskeletal organization further indicates enhanced cellular remodeling, proteostatic adaptation, and stress-responsive signaling (*Figure 4C and D*). Collectively, this co-regulated, opposing signature transitions muscle toward a fasting-adapted phenotype optimized for lipid utilization, cellular maintenance, and catabolic efficiency while actively suppressing growth and glycolytic pathways.

Proteomic and transcriptomic integration in the cerebral cortex reveals a coordinated program of remodeling in response to IF. This response is characterized by a distinct biphasic signature that balances adaptive neuroprotection with the suppression of specific signaling and structural pathways. The upregulated network points toward enhanced cellular stress resilience and metabolic adaptation. Key induced factors include the RNA-binding proteins Rbm15 and Cirbp, which regulate RNA stability and processing under stress (*Chen et al., 2018b*; *Xie et al., 2019*), alongside Txnip, a modulator of redox balance and glucose metabolism (*Deng et al., 2023*). The significant upregulation of Angptl4 suggests a shift in lipid handling, while increases in cytoskeletal-associated Ahnak and the ubiquitin ligase Rnf141 indicate active restructuring of cellular architecture and protein homeostasis (*Deng et al., 2009*; *Sundararaj et al., 2021*; *Figure 4E and F*). Conversely, a coherently downregulated module reflects an IF-induced attenuation of specific neuronal and inflammatory processes. This includes suppression of DNA replication-associated helicases (Hells), intracellular

signaling mediators (Prkg1, Gpr183), and synaptic adhesion molecules (Nrcam). Reductions in endosomal-lysosomal trafficking components and chromatin-associated factors further delineate a broad dampening of membrane trafficking, chromatin dynamics, and non-visual phototransduction. The downregulation of Serpina1e, a serine protease inhibitor, may indicate modulation of inflammatory cascades (*Figure 4E and F*). Collectively, this suggests the cerebral cortex enters an IF-adapted state that prioritizes metabolic efficiency, stress adaptation, and synaptic recalibration, while reducing energy expenditure on proliferation, excessive signaling, and constitutive maintenance pathways.

The molecular adaptations to IF reported here should be interpreted within the context of our experimental model. The dietary intervention was initiated in young, male mice. This design was chosen to model the early adoption of IF in a controlled genetic background, minimizing confounding variables introduced by age-related metabolic decline. However, initiating IF at 6 weeks of age may coincide with late developmental maturation, and this timing could influence some of the physiological and molecular responses observed. Future studies in aged populations and both sexes will be crucial to determine the translational relevance of these findings across the lifespan. Furthermore, the selection of tissues for analysis was based on their central roles in systemic metabolism, fuel utilization, and as primary targets of dietary interventions like IF. While the integration of transcriptomic and proteomic data revealed that only a fraction of expressed genes matched with DEPs, this observation is not unexpected. Our proteomic analysis identified up to a few thousand proteins, whereas transcriptomic profiling detected over 20,000 genes, highlighting the inherent disparity in the depth and coverage of these two approaches. Additionally, post-transcriptional regulatory mechanisms, such as mRNA stability, translational efficiency, and protein degradation, likely contribute to this discrepancy. This phenomenon, often referred to as 'post-transcriptional buffering', means that not all transcripts are translated into proteins, and the correlation between mRNA and protein levels is often modest. These findings underscore the complexity of gene expression regulation and emphasize the importance of multi-omics integration to fully capture the molecular adaptations induced by IF.

## Conclusion

This comprehensive study elucidates shared and tissue-specific IF-triggered metabolic adaptations across the liver, muscle, and cerebral cortex. Through integrated proteomic and transcriptomic analyses, we identified both shared and organ-specific responses to IF, revealing a complex interplay of molecular mechanisms that optimize energy utilization, enhance metabolic flexibility, and promote cellular resilience under nutrient scarcity. In the liver, IF caused upregulation of pathways involved in gluconeogenesis, fatty acid oxidation, and ketogenesis, while downregulating biosynthetic processes such as steroid hormone synthesis and cholesterol metabolism. These changes underscore the liver's central role in maintaining systemic energy homeostasis during fasting. Similarly, skeletal muscle exhibited enhanced metabolic efficiency through the upregulation of pathways like pyruvate metabolism, fatty acid biosynthesis, and AMPK signaling, while downregulating oxidative phosphorylation and thermogenesis, reflecting a shift toward alternative energy-generating processes. In the cerebral cortex, IF induced adaptive changes in autophagy, PPAR signaling, and metabolic pathways, while suppressing stress-related and biosynthetic processes, highlighting the brain's unique response to fasting.

Notably, our findings reveal that while some molecular responses to IF are conserved across tissues, such as the upregulation of Serpin A1c in the liver and its downregulation in muscle and cortex, others are highly tissue-specific. This divergence underscores the importance of organ-specific metabolic reprogramming in mediating the systemic effects of IF. Furthermore, the identification of synaptic vesicle localization as a shared biological process across all three tissues suggests a novel, unifying mechanism linking metabolic changes to cellular communication and stress adaptation. Despite the observed discordance between transcriptomic and proteomic data, attributed to post-transcriptional buffering and the inherent limitations of proteomic coverage, our multi-omics approach provides a robust framework for understanding the molecular underpinnings of IF. These insights not only advance our understanding of the physiological adaptations to fasting but also highlight potential therapeutic targets for metabolic disorders, neurodegenerative diseases, and aging. Future studies should explore the functional roles of key proteins and pathways identified here, as well as the long-term implications of IF on tissue health and systemic metabolism.

## Materials and methods

### Experimental design and animal procedures

The in vivo experiments described here received ethical approval from the Animal Care and Use Committee at La Trobe University (Approval ID: AEC21047) and adhered to the standards set by the Australian Code for the Care and Use of Animals for Scientific Purposes (8th edition) as well as the NIH Guide for the Care and Use of Laboratory Animals. Measures were taken to limit animal suffering and optimize the number of animals involved. The study complied with the ARRIVE guidelines throughout all stages of the research. Male C57BL/6 mice, aged 4 weeks, were sourced from the Animal Resources Centre in Australia and maintained at La Trobe University's animal housing facility. They were kept under a 12 hr light/12 hr dark schedule (lights on from 07:00 to 19:00) and fed a standard diet containing 20% crude protein, 5% crude fat, 6% crude fiber, 0.5% added salt, 0.8% calcium, and 0.45% phosphorus (Ridley, Victoria, Australia), with unrestricted access to water. At 6 weeks of age, the mice were randomly divided into two dietary groups: one undergoing intermittent fasting for 16 hr daily from 16:00 to 8:00 (IF) and a control group with unlimited food access (ad libitum, AL; N=8 in each experimental group). Body weight was routinely measured, and blood levels of glucose and ketones were evaluated using the FreeStyle Optium Neo device along with corresponding FreeStyle Optium test strips for glucose and ketones (Abbott Laboratories, Illinois, USA), measured after a 6 hr fast. In addition, cholesterol levels were determined using the Roche Cobas b 101 Lipid system (Cat. No. ROC06380115190, Roche Diagnostics, Basel, Switzerland), while HbA1c concentrations were assessed with the Roche Cobas b 101 HbA1c (Cat. No. ROC08038694190) kits (Roche Diagnostics, Basel, Switzerland). After 4 months on their respective diets, mice were euthanized by $CO_2$ inhalation between 07:00 and 12:00. This timing ensured that the IF group was sampled at the end of their scheduled fasting window. Immediately afterwards, the mice were perfused with chilled PBS, and tissue samples were quickly frozen in liquid nitrogen before being stored at –80 °C for subsequent analysis.

### Tissue lysate preparation

Tissue samples (30–50 mg) were lysed in 250 µl lysis buffer 6 M guanidine (Merck, Victoria, Australia), 100 mM Tris (Merck, Victoria, Australia), 10 mM TCEP (Merck, Victoria, Australia), 40 mM 2-chloroacetamide (Merck, Victoria, Australia) pH 8.5, then homogenized, sonicated, and heated at 95 °C for 5 min. Then samples were centrifuged at 20,000 × $g$ for 30 min at 4 °C and the supernatant was collected. BCA assay (Thermo Fisher Scientific, Waltham, MA, USA) was performed according to the kit's instructions to determine the protein concentration, then the samples were diluted and used for further analysis.

### Mass spectrometry (MS)-based proteomics analysis

MS-based proteomics analysis was carried out by the La Trobe University-Proteomics and Metabolomics Platform using their established workflows. Briefly, the single pot, solid phase, sample preparation strategy (*Hughes et al., 2019*) was used to clean up alkylated lysates before digestion. Proteins were captured on carboxylate-modified magnetic SpeedBeads (Cytiva 65152105050250 and 45152105050250) in a 50% ethanol environment by incubating at 24 °C for 5 min with shaking (900 rpm). Using a magnetic rack for separations, the beads were isolated and the supernatant discarded. The beads were washed three times with 80% ethanol, discarding the supernatant each time. Proteins were digested with trypsin (1:20 enzyme:protein ratio) added in 20 mM ammonium bicarbonate and incubated at 37 °C overnight on a shaker. The peptide solution was separated from the beads and collected into fresh tubes. Digested samples were acidified by addition of trifluoroacetic acid to 0.5% (v/v). The tryptic peptides were then desalted using the StageTip method (*Rappsilber et al., 2007*) and dried in a Speedvac.

### LC-MS analysis of peptides

LC-MS was performed on a Thermo Ultimate 3000 RSLCnano UHPLC system and a Thermo Orbitrap Eclipse Tribrid mass spectrometer (Thermo-Fisher Scientific, Waltham, MA, USA) at the La Trobe University-Proteomics and Metabolomics Platform. Peptides were reconstituted in 0.1% (v/v) trifluoroacetic acid (TFA) and 2% (v/v) acetonitrile (ACN), and 500 ng of peptides were loaded onto a PepMap C18 5 µm 1 cm trapping cartridge (Thermo Fisher Scientific, Waltham, MA, USA) at 12 µL/

min for 6 min before switching the trap in-line with the analytical column (nanoEase M/Z Peptide BEH C18 Column, 1.7 µm, 130 Å and 75 µm ID ×25 cm; Waters). The column compartment was held at 55 °C for the entire analysis. The separation of peptides was performed at 250 nL/min using a linear ACN gradient of buffer A (0.1% (v/v) formic acid, 2% (v/v) ACN) and buffer B (0.1% (v/v) formic acid, 80% (v/v) ACN), starting at 14% buffer B to 35% over 90 min, then rising to 50% buffer B over 15 min followed by 95% buffer B in 5 min. The column was then cleaned for 5 min at 95% buffer B and afterward ramped down to 1% buffer B over 2 min and then held at 1% buffer B for a final 3 min.

Mass spectra were collected on the Thermo Orbitrap Eclipse in Data Independent Acquisition (DIA) mode on the Orbitrap for both MS1 or "master scan" spectra and HCD DIA spectra. MS1 "master" scan parameters were: scan range of 350–1400 m/z, 120,000 resolution, max injection time 50ms, AGC target 1e6 (250%). DIA spectra were collected with a target cycle time of 3 s, resolution of 30 K, isolation window of 15, and 45 windows covering precursor range of 361–1036 m/z, and scan range of 197–2000. AGC target set to 1e6 (2000%), max IT of 55ms, HCD collision energy set to 30%. Internal mass correction was performed with the Thermo EASY-IC option.

## Database search

The mass-spectral database search was conducted using DIA-NN v181 (*Demichev et al., 2020*), with label-free DIA matching against the *Mus musculus* reference proteome downloaded from Uniprot (UP000000589_10090, 1 protein sequence per gene, March 2023). A spectral library was generated with deep learning in DIA-NN from the fasta file and library precursors reannotated from the fasta database. Min-max fragment size was 200 m/z to 1800 m/z, N-term methionine excision enabled, min-max peptide length was 7–30, min-max precursor m/z was 300–1800, precursor charge range 1–4, digest trypsin/P, missed cleavages set to 2. Fixed peptide modifications were carbamidomethyl of cysteine, and variable mods were acetylation of protein N-terminus and oxidation of methionine, and one variable mod per peptide was allowed. Processing parameters turned on were: 'use quant', 'peptidoforms', 'reanalyse', 'relaxed protein inference', and 'smart profiling'. Mass tolerances were determined from the dataset. Protein validation FDR threshold was 0.01.

## Statistical and differential expression analysis of proteomics data

Processed MS data (by the La Trobe University-Proteomics and Metabolomics Platform) were subjected to downstream statistical and bioinformatics analysis using JG lab established pipelines. For these analyses, the label-free intensities of all identified proteins were log-transformed, followed by median normalization across samples using the normalizeMedianValues function in limma R package (*Ritchie et al., 2015*). The datasets were filtered for 50% valid values across all samples (proteins with >50% missing values were excluded from downstream statistical analysis), with the remaining missing values imputed as described by *Tyanova et al., 2016*. All preprocessing steps were conducted within the publicly available FlexStat web application (*De Silva et al., 2024*). For differential protein expression analysis in response to IF, the limma package employed within the FlexStat web application was used. Pairwise comparisons against the AL control group were performed, with statistical significance assessed using the Benjamini-Hochberg procedure to control the false discovery rate (FDR), applying an adjusted p-value threshold of <0.05.

## Heat map generation and functional enrichment analyses

Heatmaps were generated using the normalized protein intensities in R with the pheatmap package (*pheatmap and Heatmaps, 2018*) with default parameters. DEPs were clustered to highlight expression patterns across experimental conditions. To infer potential biological functions associated with protein expression changes, Gene Ontology (GO) and Kyoto Encyclopedia of Genes and Genomes (KEGG) pathway enrichment analyses were performed. GO and KEGG enrichment analyses were conducted using the clusterProfiler R package (*Yu et al., 2012*), evaluating biological processes, cellular components, and molecular functions. GO terms and KEGG pathways with a q-value<0.05 were considered significantly enriched among DEPs, indicating potential functional relevance. To explore functional networks across different IF regimens, gene set enrichment analysis (GSEA) was performed using GSEA (version 4.3.3) desktop application on the proteomic dataset with curated mouse GO biological processes and pathway databases (version 2023.2; *Merico et al., 2010*). The FDR was controlled at 0.05. Enrichment scores were assigned to each functional group, where positive scores approaching

+1 indicated significant enrichment and negative scores approaching –1 indicated depletion (*Subramanian et al., 2005*). Pairwise comparisons of enrichment scores between experimental groups were evaluated using permutation testing. Based on gene set similarity (p-value <0.01), EnrichmentMap plugin (*Merico et al., 2010*; version 3.5.0), implemented in Cytoscape (version 3.10), was used to visualize overlapping gene set clusters, enabling network-based interpretation of enriched pathways.

## Protein-protein interaction network analyses

Functional connectivity among significantly altered proteins was assessed using protein-protein interaction (PPI) data curated from the STRING database (*Szklarczyk et al., 2023*), which integrates experimental evidence, computational predictions, and text mining. An IF-altered functional network for each organ was constructed to encompass interactions relevant to metabolic, regulatory, and signaling pathways. Redundant interactions were removed to minimize network complexity, and IF-specific alterations in protein expression for each organ were incorporated to generate time-resolved functional networks. Only interactions with a confidence score >0.8 were included. Network visualization was performed using the STRING plugin (version 2.2.0) implemented in Cytoscape (*Doncheva et al., 2019*). Venn diagrams were generated to identify overlapping proteins, DEPs across different IF regimens and corresponding enriched GO terms using the InteractiVenn web application (*Heberle et al., 2015*).

## Total RNA extraction and quality control for transcriptome analysis

TRIzol RNA Isolation kit (Thermo Fisher Scientific, Waltham, MA, USA) was used according to the kit's instructions to extract total RNA from liver, muscle, and cerebral cortex samples. To assess the quality of the extracted total RNA, RNA purity was determined using Nanodrop ND-1000 (Thermo Fisher Scientific, Waltham, MA, USA). Enriched RNA was of high quality, demonstrating an OD260/OD280 ratio of 1.9–2.0 from Nanodrop readings.

## cDNA library preparation and RNA sequencing

NEBNext Ultra II RNA Library Prep Kit for Illumina (New England BioLabs, MA, USA) and NEBNext Ultra II Directional RNA Library Prep Kit (New England BioLabs, MA, USA) were used according to the kit's instructions for Eukaryotic RNA-seq. Poly-T oligo-attached magnetic beads were used to purify the mRNA from total RNA. Then, a fragmentation buffer was added to the mRNA. First-strand cDNA was subsequently synthesized using random hexamer primer. Next, second-strand cDNA synthesis was performed using dUTP for directional library. After end repair, A-tailing, adapter ligation, size selection, USER enzyme digestion, amplification, and purification, the library preparation was done. The library was checked with Qubit and real-time PCR for quantification and bioanalyzer for size distribution detection. Quantified libraries were pooled and sequenced on Illumina platforms, according to effective library concentration and data amount.

## Transcriptome data mapping and differential expression analysis

The RNA sequencing results from the Illumina NovaSeq 6000 Sequencing System were output as quality and color space fasta files. The files were mapped to the Ensembl-released mouse genome sequence and annotation (*Mus musculus*; GRCm39/mm39). Hisat2 version 2.0.5 (*Kim et al., 2019b*) was used to build an index of reference genome and align paired-end clean 1 reads to the reference genome. FeatureCounts version 1.5.0-p3 (*Liao et al., 2014*) was used for the quantification of gene expression levels to count the read numbers mapped of each gene. Then FPKMs of each gene were calculated based on the length of the gene and reads count mapped to the same gene. DESeq2 R Package version 1.10.1 (*Love et al., 2014*) was used to perform differential expression analysis. Genes with p-value <0.05 found by DESeq2 were assigned as differentially expressed.

## Integrative analysis of proteome and transcriptome data

To identify robust molecular signatures driving the separation between IF and AL groups across transcriptomic and proteomic layers, supervised multi-omics integration was performed using Data Integration Analysis for Biomarker discovery using Latent variable approaches (DIABLO), implemented in the mixOmics R package version 6.28.0 (*Rohart et al., 2017*). Prior to integration, RNA-seq count data were normalized using Variance Stabilizing Transformation (VST) via the DESeq2 package (*Love*

*et al., 2014*), while proteomic label-free intensities were log-transformed. Features with near-zero variance were removed using the nearZeroVar function. A supervised N-integration block sparse PLS-DA (block.splsda) model was constructed with a design matrix set to 0.1 to prioritize discrimination between phenotypic groups while allowing for correlations between omics layers. The optimal number of features to retain for the first latent component was determined using 10-repeated seven-fold cross-validation (M-fold CV). A grid search was performed to select the feature combination minimizing the Balanced Error Rate (BER). Model performance was evaluated using the Area Under the Receiver Operating Characteristic (AUROC) curve.

## Interaction network construction and visualization

To visualize the functional connectivity between the selected transcriptomic and proteomic signatures, relevance networks were generated based on the similarity matrix inferred by the DIABLO model. Only correlations between features with an absolute coefficient >0.7 were retained to ensure high-confidence interactions. Network visualization and topology analysis were performed using the igraph R package (*Csárdi and Nepusz, 2006*). Nodes were distinctively colored to represent molecular layers, and edges were colored based on the direction of regulation.

## Functional enrichment of integrated signatures

Biological interpretation of the DIABLO-selected multi-omics signature was performed using the clusterProfiler R package version 3.8.1 (*Wu et al., 2021*). Gene and protein identifiers were converted to Entrez IDs using the org.Mm.eg.db database. Given the stringent feature selection imposed by the sparse PLS-DA model, an exploratory trend analysis was employed to identify biological trends. Enrichment analysis was conducted for Gene Ontology (GO) Biological Processes and Kyoto Encyclopedia of Genes and Genomes (KEGG) pathways. Pathways were ranked based on adjusted p-values and Fold Enrichment scores, and the top 10 most biologically relevant terms associated with the integrated signature were visualized using bubble plots.

## Immunohistochemistry

Following transcardiac perfusion fixation, liver tissue was dissected and cryoprotected overnight in 30% (w/v) sucrose in phosphate-buffered saline (PBS). Horizontal cryostat sections (14 µm thickness) were prepared and mounted on slides. For immunofluorescence staining, sections underwent antigen retrieval followed by incubation with a polyclonal anti-$\alpha$–1 antitrypsin primary antibody (PA5-122246, 1:50 dilution). Signal detection was performed using an Alexa Fluor 488-conjugated secondary antibody (ab150081). Images were acquired on a Zeiss LSM 780 confocal microscope and processed using *Cámara et al., 2011* Blue Edition software (Carl Zeiss AG).

## Statistics

For physiological measurements such as body weight and bodyweight change, values are indicated as mean ± SEM (n=8 mice/group). Significance is based on two-way ANOVA with Tukey's post-hoc test. p-value <0.05 was considered statistically significant. For physiological measurements, such as fasting blood glucose and ketone levels, values are indicated as mean ± SEM (n=8 mice/group). Significance is based on a two-tailed paired T-test. p-value <0.05 was considered statistically significant. The statistics for these comparisons were conducted using the GraphPad Prism software programme. For proteome analyses, protein validation FDR threshold was 0.01. Pairwise comparisons against the AL control group were performed, with statistical significance assessed using the Benjamini-Hochberg procedure to control the false discovery rate (FDR), applying an adjusted p-value threshold of <0.05. For function enrichment analysis, GO terms and KEGG pathways with a q-value <0.05 were considered significantly enriched among DEPs, indicating potential functional relevance. For gene set enrichment analysis (GSEA), the proteomic dataset with curated mouse GO biological processes and pathway databases FDR was controlled at 0.05. Gene set similarity was set as p-value <0.01, and the EnrichmentMap plugin (v3.5.0) in Cytoscape (v3.10) was used to visualize overlapping gene sets and interpret enriched pathways. Network visualization was performed using the STRING plugin (version 2.2.0) implemented in Cytoscape with a confidence score >0.8. For transcriptome data, differential expression analysis of the quantified transcripts was performed using DESeq2 R Package v.1.20.1. Those

transcripts with p-value <0.05 were considered to be significantly differentially expressed among the different categories.

## Acknowledgements

This work was supported by La Trobe University (start-up grant to Thiruma V Arumugam), the National Health and Medical Research Council of Australia (Grant Identification Number 2019100), Institute of Molecular and Cell Biology, Agency for Science, Technology and Research, Singapore. S D is funded by the SINGA (Singapore International Graduate Award) fellowship. *Figure 1* in this article was created using BioRender.

## Additional information

### Competing interests

The authors declare that no competing interests exist.

### Funding

| Funder | Grant reference number | Author |
| --- | --- | --- |
| National Health and Medical Research Council | 2019100 | Thiruma V Arumugam |
| Singapore international graduate award | | Senuri De Silva |

The funders had no role in study design, data collection and interpretation, or the decision to submit the work for publication.

### Author contributions

Yibo Fan, Conceptualization, Software, Formal analysis, Investigation, Methodology, Writing – original draft, Writing – review and editing; Senuri De Silva, Software, Formal analysis, Visualization, Methodology, Writing – review and editing; Nishat I Tabassum, Investigation, Methodology, Writing – review and editing; Xiangyuan Peng, Formal analysis, Methodology, Writing – review and editing; Vernise JT Lim, Validation, Investigation, Methodology, Writing – review and editing; Xiangru Cheng, Investigation, Writing – review and editing; Keshava K Datta, Resources, Data curation, Formal analysis, Validation, Visualization, Methodology, Writing – review and editing; Rohan Lowe, Resources, Formal analysis, Methodology, Writing – review and editing; Terrance G Johns, Suresh Mathivanan, Christopher G Sobey, Eitan Okun, Yong U Liu, Guobing Chen, Mitchell Kim Peng Lai, Resources, Writing – review and editing; Mark P Mattson, Dong-Gyu Jo, Conceptualization, Resources, Writing – review and editing; Jayantha Gunaratne, Conceptualization, Resources, Formal analysis, Investigation, Methodology, Writing – review and editing; Thiruma V Arumugam, Conceptualization, Resources, Software, Formal analysis, Supervision, Funding acquisition, Validation, Investigation, Visualization, Methodology, Writing – original draft, Project administration, Writing – review and editing

### Author ORCIDs

Nishat I Tabassum ⓘ https://orcid.org/0009-0004-8388-5074
Terrance G Johns ⓘ https://orcid.org/0000-0002-8874-4543
Christopher G Sobey ⓘ https://orcid.org/0000-0001-6525-9097
Eitan Okun ⓘ https://orcid.org/0000-0001-8474-1487
Mitchell Kim Peng Lai ⓘ https://orcid.org/0000-0001-7685-1424
Jayantha Gunaratne ⓘ https://orcid.org/0000-0002-5377-6537
Thiruma V Arumugam ⓘ https://orcid.org/0000-0002-3377-0939

### Ethics

All in vivo experimental procedures were approved by La Trobe University (Ethics approval number: AEC21047) Animal Care and Use Committees and performed according to the guidelines set forth by the Australian Code for the Care and Use of Animals for Scientific Purposes (8th edition) and confirmed NIH Guide for the Care and Use of Laboratory Animals.

Reviewer #1 (Public review): https://doi.org/10.7554/eLife.107332.3.sa1
Reviewer #2 (Public review): https://doi.org/10.7554/eLife.107332.3.sa2
Reviewer #3 (Public review): https://doi.org/10.7554/eLife.107332.3.sa3
Author response https://doi.org/10.7554/eLife.107332.3.sa4

## Additional files

### Supplementary files

Supplementary file 1. Raw Datasets, Differential Protein Expression Profiles, and GO Pathway Enrichment for Liver, Muscle, and Cerebral Cortex Tissues.

Supplementary file 2. KEGG Pathway Enrichment and STRING Protein–Protein Interaction Data for Liver, Muscle, and Cerebral Cortex.

Supplementary file 3. Proteins Shared Between Organs: Consolidated Dataset.

MDAR checklist

### Data availability

The mass spectrometry proteomics data have been deposited to the ProteomeXchange Consortium via the PRIDE partner repository with the dataset identifier PXD063124. High throughput RNA sequencing data from this manuscript have been submitted to the NCBI Sequence Read Archive (SRA) under accession number GSE290224.

The following datasets were generated:

| Author(s) | Year | Dataset title | Dataset URL | Database and Identifier |
|---|---|---|---|---|
| Lowe R, Arumugam T | 2026 | Intermittent Fasting Drives Metabolic Reprogramming Through Tissue-Specific and Shared Molecular Adaptations | https://www.ebi.ac.uk/pride/archive/projects/PXD063124 | PRIDE, PXD063124 |
| Xiangyuan YF, Arumugam T | 2026 | Intermittent Fasting Reprograms Chromatin Accessibility to Modulate Gene Expression in Brain and Muscle | https://www.ncbi.nlm.nih.gov/geo/query/acc.cgi?acc=GSE290224 | NCBI Gene Expression Omnibus, GSE290224 |

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
