## [Editor Report · eLife Assessment]

This is a **solid** paper on intermittent fasting that will be of interest to readers. The data presented are certainly **valuable** as a resource. The findings of both shared and tissue-specific signatures, both at the proteomic and transcriptomic levels, align well with what has been established and bring new insight into metabolic adaptation and its consequences in muscle, cortex, and liver. The organ specific changes unveiled by proteomics in response to IF reveal unique rewiring of metabolic, signaling and physiological function.

---

## [Referee Report · Reviewer #1 (Public review)]

Summary:

In this study, authors employed comprehensive proteomics and transcriptomics analysis to investigate the systemic and organ-specific adaptations to IF in male and they found that shared biological signaling processes were identified across tissues, suggesting unifying mechanisms linking metabolic changes to cellular communication, which reveal both conserved and tissue-specific responses by which IF may optimize energy utilization, enhance metabolic flexibility, and promote cellular.

Strengths:

This study detected multiple organs including liver, brain and muscle and revealed both conserved and tissue-specific responses to IF.

Weaknesses:

(1) Why did the authors choose liver, brain and muscle but not other organs such as heart and kidney? The latter are proven to be the large consumer of ketones, which is also changed in the IF treatment of this study.

(2) The proteomics and transcriptomics analysis were only performed at 4 months. However, a strong correlation between IF and the molecular adaptions should be time points-dependent.

(3) The context lack section of "discussion", which shows the significance and weakness of the study.

(4) There is no confirmation for the proteomic and transcriptomic profiling. For example, the important changes in proteomics could be further identified by a Western blot.

---

## [Referee Report · Reviewer #2 (Public review)]

Summary:

Fan and colleagues measure proteomics and transcriptomics in 3 organs (liver, skeletal muscle, cerebral cortex) from male C57BL/6 mice to investigate whether intermittent fasting (IF; 16h daily fasting over 4 months) produces systemic and organ-specific adaptations.

They find shared signaling pathways, certain metabolic changes and organ-specific responses that suggest IF might affect energy utilization, metabolic flexibility while promoting resilience at the cellular level.

Strengths:

The fact that there are 3 organs and 2 -omics approaches is a strength of this study.

Weaknesses:

Poor figures presentation and knowledge of the literature. One sex (male).

On resubmission the Authors' decision to discriminate the organ-specific from the organ-shared effects of intermittent fasting (IF) also enabled them to more precisely determine the lack of correspondence between transcriptomics and proteomics, i.e., not all transcripts lead to protein translation.

---

## [Referee Report · Reviewer #3 (Public review)]

Summary:

Fan et al utilize large omics data sets to give an overview of proteomic and gene expression changes after 4 moths of intermittent fasting (IF) in liver, muscle and brain tissue. They describe common and district pathways altered under IF across tissues using different analysis approaches. Main conclusions presented are the variability in responses across tissues with IF. Some common pathways were observed, but there were notable distinctions between tissues.

Strengths:

(1) The IF study was well conducted and ran out to 4 months which was a nice long-term design.

(2) The multi omics approach was solid and additional integrative analysis was complementary to the illustrate the differential pathways and interactions across tissues.

(3) The authors did not over-step their conclusions and imply an overreached mechanism.

Weaknesses:

The weaknesses, which are minor, include use of only male mice and the early start (6 weeks) of the IF treatment. However, the authors have provided justification on why they chose male mice and the time points used in the study.

---

## [Author Response]

The following is the authors’ response to the original reviews.

**Public Reviews:**

**Reviewer #1 (Public review):**
Summary:In this study, the authors employed comprehensive proteomics and transcriptomics analysis to investigate the systemic and organ-specific adaptations to IF in males. They found that shared biological signaling processes were identified across tissues, suggesting unifying mechanisms linking metabolic changes to cellular communication, which revealed both conserved and tissue-specific responses by which IF may optimize energy utilization, enhance metabolic flexibility, and promote cellular resilience.Strengths:This study detected multiple organs, including the liver, brain, and muscle, and revealed both conserved and tissue-specific responses to IF.

We appreciate the recognition of the study’s strengths and the opportunity to clarify the points raised.

Weaknesses:(1) Why did the authors choose the liver, brain, and muscle, but not other organs such as the heart and kidney? The latter are proven to be the largest consumers of ketones, which is also changed in the IF treatment of this study.

We agree that the heart and kidney are critical organs in ketone metabolism. Our selection of the liver, brain, and muscle was guided by their distinct metabolic functions and relevance to systemic energy balance, neuroplasticity, and locomotor activity, key domains influenced by intermittent fasting (IF). These tissues also offer complementary perspectives on central and peripheral adaptations to IF. Notably, we have previously examined the effects of IF on the heart (eLife 12:RP89214), and we fully acknowledge the importance of the kidney. We intend to include it in future studies to broaden the scope and deepen our understanding of IF-induced systemic responses.

(2) The proteomics and transcriptomics analyses were only performed at 4 months. However, a strong correlation between IF and the molecular adaptations should be time point-dependent.

We appreciate this insightful comment. The 4-month time point was selected to capture long-term adaptations to IF, beyond acute or transitional effects. While we acknowledge that molecular responses to IF are time-dependent, our goal in this study was to establish a foundational understanding of sustained systemic and tissue-specific changes. We fully agree that a longitudinal approach would provide deeper insights into the temporal dynamics of IF-induced adaptations. To address this, we are currently undertaking a comprehensive 2-year study that is specifically designed to explore these time-dependent effects in greater detail.

(3) The context lacks a "discussion" section, which would detail the significance and weaknesses of the study.

We appreciate this observation. The manuscript was originally structured to emphasize results and interpretation within each section, but we recognize that a dedicated discussion section would enhance clarity and contextual depth. In the revised version, we will add a comprehensive discussion section addressing broader implications, limitations, and future directions of the study.

(4) There is no confirmation for the proteomic and transcriptomic profiling. For example, the important changes in proteomics could be further identified by a Western blot.

We acknowledge the importance of orthogonal validation to support high-throughput findings. While our study primarily focused on uncovering systemic patterns through proteomic and transcriptomic profiling, we agree that targeted confirmation would strengthen the conclusions. To this end, we have included immunohistochemical validation of a key protein common to all three organs— Serpin A1C. Additionally, we are planning a dedicated follow-up study to expand functional validation of several key proteins identified in this manuscript, which will be pursued as a separate project.

**Reviewer #2 (Public review):**
Summary:Fan and colleagues measure proteomics and transcriptomics in 3 organs (liver, skeletal muscle, cerebral cortex) from male C57BL/6 mice to investigate whether intermittent fasting (IF; 16h daily fasting over 4 months) produces systemic and organ-specific adaptations.They find shared signaling pathways, certain metabolic changes, and organ-specific responses that suggest IF might affect energy utilization, metabolic flexibility, while promoting resilience at the cellular level.Strengths:The fact that there are 3 organs and 2 -omics approaches is a strength of this study.

We appreciate the reviewer’s recognition of the breadth of our study design. By integrating proteomics and transcriptomics across three metabolically distinct organs, we aimed to provide a comprehensive view of systemic and tissue-specific adaptations to IF. This multi-organ, multi-omics approach was central to uncovering both conserved and divergent biological responses.

Weaknesses:(1) The analytical approach of the data generated by the present study is not well posed, because it doesn't help to answer key questions implicit in the experimental design. Consequently, the paper, as it is for now, reads as a mere description of results and not a response to specific questions.

We thank the reviewer for this important observation. Our initial aim was to establish a foundational atlas of molecular changes induced by IF across key organs. However, we recognize that clearer framing of the biological questions would enhance interpretability. In the revised manuscript, we will have restructured the introduction, results, and discussion to align more explicitly with specific hypotheses, particularly those related to energy metabolism, cellular resilience, and inter-organ signaling. We have also added targeted analyses and clarified how each dataset contributes to answering these questions.

(2) The presentation of the figures, the knowledge of the literature, and the inclusion of only one sex (male) are all weaknesses.

We appreciate this feedback and agree that these are important considerations. Regarding figure presentation, we will revise several figures for improved clarity, add more descriptive legends, and reorganize supplemental materials to better support the main findings. On the literature front, we will expand the discussion to include recent and relevant studies on IF, metabolic adaptation, and sex-specific responses. As for the use of only male mice, this was a deliberate choice to reduce hormonal variability and focus on establishing baseline molecular responses. We fully acknowledge the importance of sex as a biological variable and will soon be conducting studies in female mice to address this gap.

**Reviewer #3 (Public review):**
Summary:Fan et al utilize large omics data sets to give an overview of proteomic and gene expression changes after 4 months of intermittent fasting (IF) in liver, muscle, and brain tissue. They describe common and distinct pathways altered under IF across tissues using different analysis approaches. The main conclusions presented are the variability in responses across tissues with IF. Some common pathways were observed, but there were notable distinctions between tissues.Strengths:(1) The IF study was well conducted and ran out to 4 months, which was a nice long-term design.(2) The multiomics approach was solid, and additional integrative analysis was complementary to illustrate the differential pathways and interactions across tissues.(3) The authors did not overstep their conclusions and imply an overreached mechanism.

We sincerely thank the reviewer for acknowledging the strengths of our study design and analytical approach. We aimed to strike a careful balance between comprehensive data generation and cautious interpretation, and we appreciate the recognition that our conclusions were appropriately framed within the scope of the data.

Weaknesses:The weaknesses, which are minor, include the use of only male mice and the early start (6 weeks) of the IF treatment. See specifics in the recommendations section.

We appreciate the reviewer’s thoughtful comments. The decision to use male mice and initiate IF at 6 weeks was based on minimizing hormonal variability and capturing early adult metabolic programming. We acknowledge that sex and developmental timing are important biological variables. To address this, we are conducting parallel studies in female mice and evaluating IF initiated at later life stages. These follow-up investigations will help determine the extent to which sex and timing influence the molecular and physiological outcomes of IF.

**Recommendations for the authors:**

**Reviewing Editor Comments:**
The editor suggested addressing points regarding the young age at diet onset, use of males only, and justification for the choice of tissues analyzed without requiring new data generation.

We agree that these are important points for context. We have now added a dedicated paragraph to the Discussion section (page 22) to explicitly acknowledge and discuss these as limitations of our study. We justify our initial experimental design choices in the context of the existing literature while acknowledging the valuable insights that studies in females and with different diet onset timings would provide.

The editor and reviewers recommended a more integrative analysis, suggesting the use of freely available tools, and a deeper discussion to frame the work against the existing literature.

We thank the editor for this excellent suggestion. In response to this and the detailed points from Reviewer #2, we have performed a new, integrated multi-omics analysis using Latent variable approaches (DIABLO), implemented in the mixOmics R package version 6.28.0 tool, a state-of-the-art, freely available package for integrative multi-omics analysis. This new analysis, presented in a new Figure 4 and described in the Results section (pages 20-23), identifies the key sources of variation across tissues and omics layers, directly addressing the request for a true integrative approach. Furthermore, we have thoroughly revised the Results and Discussion to more sharply frame our findings and highlight the new insights gleaned from our study.

The editor requested clarification on whether mice were fasted at euthanasia and to rephrase the statement on page 12 regarding mitochondrial pathways.

- We have clarified in the Methods section (page 4) that mice were euthanized at the end of their fasting period, precisely detailing the stage of the IF cycle.

- We thank the editor for this critical correction. We have rephrased the statement on page 12 to more accurately reflect that we observed a lower abundance of proteins involved in mitochondrial oxidative pathways, and we now carefully discuss the important distinction between protein abundance and functional activity in this context.

The editor noted that the introduction is missing key citations and should acknowledge foundational work.

We apologize for this oversight. We have now revised the Introduction to include several key foundational citations that were previously missing, ensuring proper credit to the important work of our colleagues.

**Reviewer #2 (Recommendations for the authors):**

We thank the reviewer for their exceptionally detailed and helpful technical suggestions, which have greatly improved the analytical rigor of our manuscript.

(1) & (4) 3D PCA and Integrated Multi-Omics Analysis:

We agree with the reviewer that a more sophisticated integrative analysis was needed. As detailed in our response to the editor, we have replaced the original side-by-side analysis with a proper integrated multi-omics analysis using Latent variable approaches (DIABLO), implemented in the mixOmics R package version 6.28.0 tool. This new analysis simultaneously models the proteomic and transcriptomic data from all three organs, identifying shared and tissue-specific sources of variation. This directly and more powerfully validates our claim of "conserved and tissue-specific responses." The results of this analysis are now central to our revised Results section and Figure 4 and supplementary figures (PCA analysis).

(2) Concordance/Discordance Analysis:

This is an excellent point. We have now performed a comprehensive analysis of transcript-protein concordance for the differentially expressed molecules in each tissue. A new figure 4 summarizes these findings, and we discuss the biological implications of both concordant and discordant pairs in the Results section.

(3) Organ-Specific Functional Remodeling:

We have taken this advice to heart. The new analysis inherently addresses whether the functional remodeling is shared or tissue-specific.

(5) Missing Citations:

We have thoroughly reviewed the literature and added key citations throughout the manuscript, particularly in the Introduction and Discussion, to properly situate our work within the field.

(6) Starting Results with Supplementary Data:

As the study design, including the timing of experimental interventions and blood and tissue collections, is summarized in the supplementary figures, the Results and Discussion section begins with those figures. However, we have now renamed the figures according to the eLife style, in which supplementary figures are linked to the main figures. This ensures a more logical and coherent flow.

(7) Figure Presentation and Explanation:

We have completely revised all figures to improve their clarity, consistency, and professional appearance. We have also carefully gone through the manuscript to ensure that every panel in every figure is explicitly mentioned and explained in the main text.

**Reviewer #3 (Recommendations for the authors):**

We thank the reviewer for their important comments regarding the model system.

(1) Sex Differences and Limitations:

We fully agree that studying sex differences is a critical and profound aspect of dietary interventions. As noted in our response to the editor, we have added a paragraph to the Discussion to explicitly acknowledge this as a key limitation of our current study. We discuss the existing evidence for sex-specific responses to IF and state that this is an essential direction for future research.

(2) Early Diet Onset and Developmental Programs:

This is a valuable point. We have added text to the Discussion acknowledging that starting IF at 6 weeks of age could potentially interact with developmental programs. We discuss this as a consideration for interpreting our data and for the design of future studies.

We believe that our revised manuscript is substantially stronger as a result of addressing these comments. We are grateful for the opportunity to improve our work and hope that you and the reviewers find these responses and revisions satisfactory.